# Reply to Auclin et al. Comment on “Hopkins et al. Value of the Lung Immune Prognostic Index in Patients with Non-Small Cell Lung Cancer Initiating First-Line Atezolizumab Combination Therapy: Subgroup Analysis of the IMPOWER150 Trial. *Cancers* 2021, *13*, 1176”

**DOI:** 10.3390/cancers13153763

**Published:** 2021-07-27

**Authors:** Ashley M. Hopkins, Ganessan Kichenadasse, Ahmad Y. Abuhelwa, Ross A. McKinnon, Andrew Rowland, Michael J. Sorich

**Affiliations:** 1College of Medicine and Public Health, Flinders University, Bedford Park, SA 5042, Australia; ganessan.kichenadasse@flinders.edu.au (G.K.); ahmad.abuhelwa@flinders.edu.au (A.Y.A.); ross.mckinnon@flinders.edu.au (R.A.M.); andrew.rowland@flinders.edu.au (A.R.); michael.sorich@flinders.edu.au (M.J.S.); 2Department of Medical Oncology, Flinders Medical Centre, Adelaide, SA 5042, Australia

We thank Auclin et al. [1] for the comments on our manuscript in *Cancers* titled “Value of the Lung Immune Prognostic Index (LIPI) in Patients with Non-Small Cell Lung Cancer (NSCLC) Initiating First-Line Atezolizumab Combination Therapy: Subgroup Analysis of the IMpower150 Trial” [2]. We read with interest the elements of the approach proposed by Auclin et al.’s [1] to integrate LIPI into routine clinical practice and, specifically: (1) to validate LIPI retrospectively in previous clinical trials with immunotherapy; (2) to design prospective clinical trials including LIPI as a stratification factor; and (3) to design prospective clinical trials using LIPI as a marker for guiding treatment selection. However, caution is required to ensure the development of ‘host-related inflammatory indices’ for immunotherapies are well planned and target the key factors enabling precision immunotherapy use in oncology.

Auclin et al. [1] provide valid comment that LIPI needs to continue to be investigated in post hoc analysis of clinical trials involving immunotherapies. It is well appreciated that single clinical trials are usually underpowered to statistically evaluate differences in treatment effect between arms across baseline subgroups [3]. Thus, it is not surprising that in our analysis of IMpower150 the change in relative overall survival treatment effect of atezolizumab-bevacizumab-carboplatin-paclitaxel (ABCP) versus bevacizumab-carboplatin-paclitaxel (BCP) (and ACP vs. BCP) did not show any statistical difference across the LIPI subgroups (P-interaction = 0.66), despite no absolute benefit in median overall survival being observed in the poor LIPI group [2]. Reiterating, research needs: (1) to pool trials to increase the power to conclusively determine treatment effects within the poor LIPI group; (2) to evaluate LIPI performance for other immune checkpoint inhibitor combination approaches; and (3) to evaluate LIPI prognostic performance within a large real-world cohort. Auclin et al. [1] also highlight EGFR/ALK and immunotherapy cross-over populations are subgroups of interest for further investigation, which were beyond the scope of our analysis in a single trial.

Based on current and emerging data [1,2,4,5,6] there is validity in the recommendation that prospective investigations of immunotherapies should consider inclusion of LIPI as a stratification factor in trial design. At a minimum, immunotherapy trial publications need to report the distributions of inflammatory markers (e.g., neutrophil to lymphocyte ratio, lactate dehydrogenase, c-reactive protein) to enable assessment of randomization validity and potential introduction of bias to trial findings [7].

With respect to LIPI guiding treatment selection in planned prospective clinical trials, careful consideration of the key clinical potentials of the inflammatory indices are required. As requested, Figure 1 provides Kaplan–Meier estimates of overall survival and progression-free survival according to LIPI groups for patients treated with BCP in IMpower150 [2]. Within the 381 evaluable participants randomised to BCP, median OS ranged from 9 months for the poor LIPI group to 23 months for the good LIPI group (*p* < 0.001) [2]. LIPI OS discrimination performance (c-statistic) was 0.67 in the BCP cohort, compared to 0.63 and 0.63 in the ABCP and ACP cohorts, respectively [2]. Such data presents LIPI is not immunotherapy specific, and that limiting immunotherapy access in clinical trials is premature. Rather, we propose that LIPI is worthy of consideration for regulators as a baseline read-out for pay for performance strategies and as an easy-to-use clinical tool for both shared decision making and setting realistic expectations of likely outcomes to immunotherapies. We also encourage the comparison of LIPI to other emerging predictor tools [7] and the incorporation of clinicopathological makers (e.g., c-reactive protein, body mass index, or concomitant medicines) which may potentiate the immunotherapy specificity of the LIPI metric [7,8,9,10,11].

In conclusion, LIPI is a significant prognostic marker of overall survival and progression-free survival in patients with chemotherapy-naïve, metastatic non-squamous NSCLC who initiate any of ABCP, ACP, or BCP therapy. With respect to treatment benefit, the study provides additional evidence that within the poor LIPI group immunotherapies are associated with reduced overall survival and progression-free survival benefit. This highlights the importance of future investigations of the clinical potential of immunotherapy prediction models to support shared decision making, provide realistic expectations to treatment, and inform regulators of cost-effectiveness across subgroups (i.e., small enhancements in prediction performance compared to LIPI have significant potential for major clinical impacts).

## Figures and Tables

**Figure 1 cancers-13-03763-f001:**
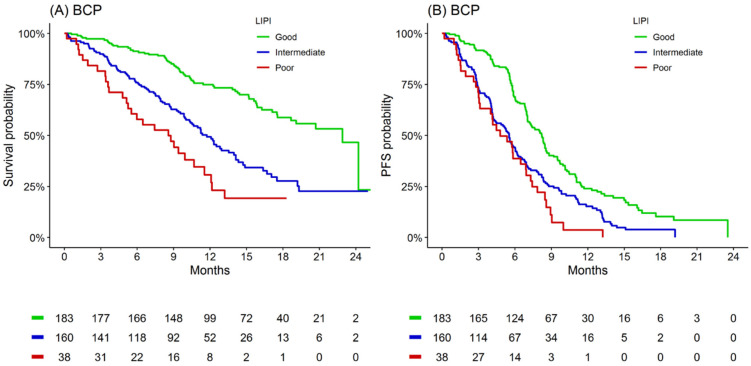
Kaplan-Meier estimates of overall survival and progression-free survival according to LIPI group for patients treated with BCP. (**A**) Overall survival and (**B**) progression-free survival.

## Data Availability

Data and results for this comment were extracted from publication Hopkins et al. Value of the Lung Immune Prognostic Index in Patients with Non-Small Cell Lung Cancer Initiating First-Line Atezolizumab Combination Therapy: Subgroup Analysis of the IMpower150 Trial. Cancers, 2021, 13(5), 1176.

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
