# Peer review of "Reply to Auclin et al. Comment on “Hopkins et al. Value of the Lung Immune Prognostic Index in Patients with Non-Small Cell Lung Cancer Initiating First-Line Atezolizumab Combination Therapy: Subgroup Analysis of the IMPOWER150 Trial. Cancers 2021, 13, 1176”"

_cancers, 2021, doi:10.3390/cancers13153763_

Round 1
Reviewer 1 Report
This is a reply to a recent letter regarding the published article entitled 'The clinical potential of the Lung Immune Prognostic Index for immunotherapy'.
It iteratively addresses the discussion points raised in the letter from Auclin and colleagues, also presenting requested data for LIPI survival estimates using the BCP regimen.
Two sentences need to be rephrased, otherwise this is suitable to publish as part of an ongoing and important discussion for this convenient immunotherapy biomarker:
- "Such data presents LIPI is not immunotherapy specific, and that limiting immunotherapy access in clinical trials is pre-empt" Do the authors mean 'premature'?
- Final sentence, "Beckoning further investigation of the clinical potential of immunotherapy prediction models to support shared decision making, provide realistic expectations to treatment, and inform regulators of cost-effectiveness across subgroups." It feels like this sentence is incomplete - rephrase.
Author Response
We thank the reviewer for their comments and appreciate that they found the reply letter to be informative. We have updated the sentences to which reviewer referred to, as they were poorly written.
Changes:
As suggested the manuscript has been updated to include:
‘Such data presents LIPI is not immunotherapy specific, and that limiting immunotherapy access in clinical trials is premature.’
The final sentence of the manuscript has been updated to:
‘This highlights the importance of future investigations of the clinical potential of immunotherapy prediction models to support shared decision making, provide realistic expectations to treatment, and inform regulators of cost-effectiveness across subgroups (i.e., small enhancements in prediction performance compared to LIPI have significant potential for major clinical impacts).’